# Risk factors for third-generation cephalosporin resistant *Enterobacteriaceae* in gestational urine cultures: A retrospective cohort study based on centralized electronic health records

Alex Guri[1,2☯]*, Natalie Flaks-Manov[3☯], Adi Ghilai[3], Moshe Hoshen[3], Orna Flidel Rimon[2,4], Pnina Ciobotaro[2,5], Oren Zimhony[2,5]

1 Division of Pediatrics, Kaplan Medical Center, Rehovot, Israel, 2 The School of Medicine, The Hebrew University and Hadassah Medical Center, Jerusalem, Israel, 3 Clalit Research Institute, Chief Physician's Office, Clalit Health Services, Tel Aviv, Israel, 4 Neonatology Department, Kaplan Medical Center, Rehovot, Israel, 5 Infectious Diseases Unit, Kaplan Medical Center, Rehovot, Israel

☯ These authors contributed equally to this work.
* alexgur@clalit.org.il, alexguri@gmail.com

## Abstract

Third-generation-cephalosporin resistant *Enterobacteriaceae* (3GCR-EB) carriage in pregnant women poses challenges for infection control and therapeutic decisions. The factors associated with multidrug resistant *Enterobacteriaceae* carriage in the gestational period are not well documented. The aim of our study was to identify risk factors associated with 3GCR-EB isolation in gestational urine cultures. The study was designed as retrospective cohort based on centralized electronic health records database. Women delivered in Clalit Health Services hospitals in Israel in 2009–2013 and provided urine culture(s) during pregnancy were included. Multivariable analysis using the Generalized Estimating Equations model was used to assess risk factors for 3GCR-EB isolation in gestational urine cultures. The study included 15,282 pregnant women with urine cultures yielding *Enterobacteriaceae* (EB). The proportion of 3GCR-EB in EB isolates was 3.9% (n = 603). The following risk factors were associated with 3GCR-EB isolation: multiple hospital admissions during the year before delivery (OR,1.47;95% CI,1.21–1.79), assisted fertilization procedure (OR,1.53; 95% CI,1.12–2.10), Arab ethnicity (OR,1.22; 95% CI,1.03–1.45), multiple antibiotic courses (OR,1.76; 95% CI,1.29–2.40), specifically, cephalosporins (OR,1.56; 95% CI,1.26–1.95), fluoroquinolones (OR,1.34; 95% CI,1.04–1.74), or nitrofurantoin (OR,1.29; 95% CI,1.02–1.64). The risk factors identified by this study for 3GCR-EB in gestation, can be easily generalized for pregnant women in the Israeli population. Moreover, these risk factors, other than ethnicity, are applicable to pregnant women worldwide. The information of previous antibiotic treatments, hospitalization in the last year and assisted fertilization procedure can be easily accessed and used for appropriate infection control practices and antimicrobial therapy.

**Data Availability Statement:** All relevant data are within the manuscript.

**Funding:** The author(s) received no specific funding for this work.

**Competing interests:** The authors have declared that no competing interests exist.

## Introduction

Enterobacteriaceae (EB) species are a well-established cause of obstetric and neonatal infections [1]. *E. coli* is one of the leading pathogens causing sepsis in the puerperium and, in one nationwide study, was found responsible for 21% of severe maternal sepsis cases in the UK [2]. Asymptomatic bacteriuria, usually caused by EB, occurs in 5–10% of all pregnancies [3]. Identifying and effectively treating asymptomatic bacteriuria and symptomatic urinary tract infections in early gestation prevents most cases of pyelonephritis and is associated with improved pregnancy outcomes [4]. Therefore, pregnant women should submit urine cultures at least once during pregnancy to detect asymptomatic bacteriuria [5]. These screening cultures are first obtained between weeks 12–15 of pregnancy. This practice is internationally accepted and the compliance was found to be high in a Swedish cohort study [6].

Third-generation cephalosporins (3GC) resistant EB (3GCR-EB) are a major threat, resulting in higher mortality rates than non-resistant strains of EB [7]. Extended-spectrum β-lactamase (ESBL) production is the principle mechanism that underlies resistance to 3GC in EB [8], and as such, some defined resistance to 3GC as a surrogate marker of organisms that produce an ESBL [9]. Extended-spectrum β-lactamase (ESBL) are β-lactamases that hydrolyse third-generation cephalosporins (3GC), aztreonam, and are inhibited by clavulanic acid [10]. ESBL-PE were originally found almost exclusively in the hospital setting [10]. A global prevalence of carrier state for ESBL-PE in the community was estimated at 14% by a recent meta-analysis of 66 studies, with significantly higher rates in studies from Southeast Asia and Africa [11]. The increased threat of 3GCR-EB is attributed largely to the emergence of Extended-spectrum β-lactamase-producing EB (ESBL-PE) strains in the community, particularly of the CTX-M type [12].

There are few reports of the prevalence of ESBL–PE in pregnancy. A surveillance study (using rectal cultures) from Argentina showed a rate of 5.4% [13]. A similar study from Madagascar identified an ESBL-PE carrier rate of 18.5% rate among pregnant women [14]. The prevalence of ESBL-PE in the general population of pregnant women in Israel is unknown. In a recently published research from south Israel, the rate of ESBL-PE in subset of women whose neonates were admitted to a NICU was 21.5% [15]. There is good evidence that isolation of ESBL-PE in urine cultures reflects faecal carriage of these organisms [16].

Several studies have highlighted the importance of maternal-neonatal transmission of resistant gram-negative bacteria for neonatal colonization and morbidity. These studies showed that an ESBL-PE infected mother is an independent risk factor for neonatal colonization [15]. Except for the studies mentioned above [13, 14], the rate and the factors associated with ESBL-PE carriage in the gestational period are not well documented. Understanding of 3GCR-EB as a surrogate for ESBL-PE epidemiology in pregnant women may guide empirical antibiotic treatment and appropriate infection control measures both to mothers and their newborns.

We aimed to identify risk factors associated with 3GC-resistant carbapenem sensitive EB isolation in urine of pregnant women.

## Methods

### Study design and population

A retrospective cohort study of women who delivered in eight Israeli hospitals of the Clalit Health Services (CHS) system between 2009 and 2013 and had performed at least one urine culture within one year prior to delivery with at least one positive urine culture for EB. The range of one year rather than nine months prior to delivery was chosen to allow inclusion of

pertinent risk factors for subsequent 3GCR-EB isolation following fertilization. We did not distinguish between isolation of EB in asymptomatic bacteriuria (the majority) and symptomatic urinary infections, as both result from prior colonization, and both require antimicrobial therapy during pregnancy.

The index date for estimation of relevant factors for subsequent isolation of 3GCR-EB was the EB positive urine culture date nearest to delivery. Therefore, for more than one positive culture, the latest culture date was chosen for further analysis. When both susceptible EB and 3GCR-EB were isolated, the date of the last 3GCR-EB positive culture before delivery was selected.

## Setting

CHS is Israel's largest integrated payer-provider healthcare system. CHS owns and operates eight general hospitals and 1,500 primary and specialty clinics throughout the country, providing healthcare services to more than four million members (52% of the population). All Israeli citizens are entitled to free health insurance coverage, and the rate of switching between healthcare organizations is low (1–1.5% per year [17]).

CHS's patient electronic health records (EHRs) have been recorded and collated in the CHS data warehouse since the early 2000s. These patient EHRs include inpatient, outpatient, demographic, administrative, and laboratory data, which are linked and available for each patient's healthcare provider(s).

## Data collection

Data of gestational urine culture results and antibiotic susceptibility profiles were obtained from the centralized CHS data warehouse. Results from routine gestational urine cultures collected in CHS outpatient clinics and analysed in microbiology laboratories were included in this study.

For research purposes, all datasets were fully anonymized before we accessed them in keeping with the standard operating procedures of CHS's Data Extraction Committee. The study was approved by CHS's Institutional Review Board.

## Identification of cases with 3GCR-EB gestational bacteriuria

Midstream urine samples were submitted to culture routinely during pregnancy follow-up visits for most of study population. These gestational urine cultures obtained in outpatient pregnancy clinics constitute the majority of the gestational cultures in this study. Other urine culture samples available were included as well to allow comprehensive assessment of all relevant gestational urine cultures.

In the CHS microbiology laboratories positive cultures were defined as the isolation of EB greater or equal to $10^4$ CFU/mL. EB species included were *Escherichia*, *Klebsiella*, *Proteus*, *Citrobacter*, *Enterobacter*, *Hafnia*, *Morganella*, *Providencia*, and *Serratia*. In some cases more than one species of EB were identified in a single culture. EB isolates were identified, and susceptibility to antimicrobials, ESBL-PE and AmpC phenotype were determined by VITEK II (card AST-N270, bioMerieux, Marcy, L'Etoile, France) and phenotypically by disks containing 30μg of cefotaxime and ceftazidime either alone or in combination with 10μg of clavulanic acid (Oxoid, Hampshire, England), according to Clinical and Laboratory Standards Institute (CLSI) criteria [18]. The data retrieval was based on the susceptibility profile of the isolates. Study enrolment included all cases that were reported as resistant to ceftriaxone and ceftazidime and susceptible to carbapenems, therefore, although each bacterium has undergone a precise identification of resistance mechanism, we use a broad definition of 3GCR-EB.

## Clinical and demographic characteristics

Demographic and clinical characteristics included: patient age of woman at the delivery date (younger than 20 years, 20–35 years, older than 35 years); socioeconomic status [SES] (low, medium, or high, according to geostatistical data provided by the Israeli Central Bureau of Statistics); ethnicity as determined by the location of the clinic at which the patient receives primary care (Jewish or Arab sector); number of foetuses per delivery (single or multiple); type of conception (spontaneous or assisted fertilization procedures [AFP] defined by ICD-9 procedures codes V26X); urinary tract anomalies (defined by ICD-9 codes 591–594, 596, 598, 599.1–9), high-risk pregnancy (ICD-9 V23, 642); maternal diabetes (ICD-9 250) and gestational diabetes (ICD-9 648.0, 648.8 or abnormal glucose tolerance test after exclusion of pregestational diabetes mellitus); maternal history of malignancy (as defined in CHS registry of chronic diseases, a compendium of any primary hospital care diagnosis, as well as the Israel National Cancer Registry); inflammatory bowel disease (ICD-9 555,556); maternal BMI (last measurement before index date); number of hospital admissions during the year before delivery (0, 1, or more than 1); antibiotics (of any kind) prescription during the year before index date (yes or not); number of dispensed antibiotic courses during the year before the index date (0, 1, 2, or more than 2); the number of different types of antibiotics prescribed (0, 1, 2, or more than 2); and type(s) of the antibiotics dispensed (tetracyclines, penicillins, cephalosporins, sulfonamides, macrolides, aminoglycosides, fluoroquinolones, nitrofurantoin or others).

## Statistical methods

Descriptive statistics were calculated for all variables that were expected to affect the primary outcome (isolation of 3GCR-EB in gestational urine culture) based on previous studies [7, 19–23] and presented as numbers and percentages. Unadjusted odds ratios (OR) and 95% confidence intervals (CI) were also calculated and presented. Univariate analyses by $\chi^2$ test were conducted for categorical variables and ANOVA tests were conducted for continuous variables. A multivariate analysis using the Generalized Estimating Equations (GEE) model [24] was used to assess whether variables found in the univariate analysis were significantly associated with 3GCR-EB isolation in gestational urine cultures. The GEE model accounts for clustering of mothers who delivered multiple times during the four years of study. All analyses were performed using R Version 3.2.2.

# Results

## Study enrolment and demographic data

The study included 134,152 women who had urine cultures performed during the year prior to delivery, of which, 15,282 (11.4%) had a positive EB culture (Fig 1). The proportion of 3GCR-EB was 3.9% (n = 603). Most women in our cohort (78.0%) were between the ages of 20 and 35 years, more than a half (53.7%) were classified as low SES and 68.2% were Jewish (Table 1).

## Bacteriologic data

As shown in Table 1, the study included 15,439 EB isolates in 15,282 positive urine cultures. Of these, 11,515 (74.6%) yielded *E. coli*, 1,473 (9.5%) yielded *K. pneumoniae* and 2,451 (15.9%) yielded other EB. Among *E. coli*, *K. pneumoniae*, and other EB species, 437 (3.8%), 90 (6.1%), and 84 (3.4%) were defined as 3GCR-EB, respectively. A detailed list of isolated EB species is presented in Table 2.

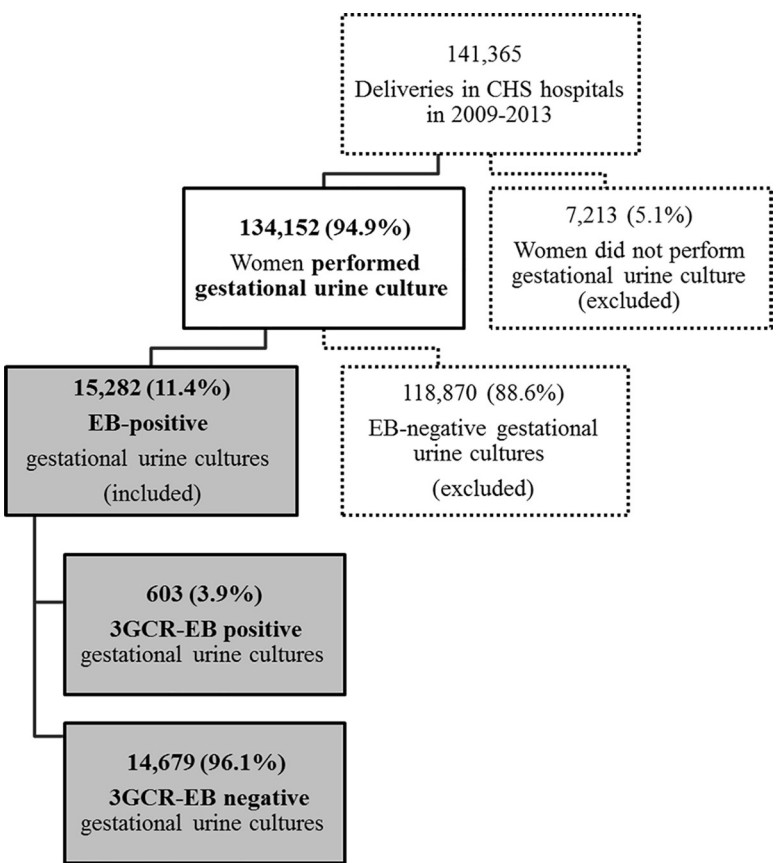

**Fig 1. Flowchart of the study population.** Abbreviations: CHS, Clalit Health Services; EB, Enterobacteriaceae; 3GCR-EB, Third-generation-cephalosporin resistant Enterobacteriaceae.

## Maternal characteristics and univariate analysis of factors associated with 3GCR-EB isolation

Several variables in the univariate analysis were significantly associated with increased risks for 3GCR-EB isolation in gestational urine culture: Arab ethnicity (p = 0.004), multiple pregnancies (p = 0.027), pregnancy achieved by AFP (p = 0.006), urinary tract anomalies (p = 0.001), hospital admissions before the hospitalization for delivery (p < 0.001), antibiotics prescription (p < 0.001), more than two dispensed antibiotic courses (p < 0.001), penicillins (p < 0.001), cephalosporins (p < 0.001), sulfonamides (p < 0.001), aminoglycosides (p < 0.001), and fluoroquinolones (p < 0.001). These variables were chosen for multivariable risk factors analyses (Table 1).

## Risk factors for 3GCR-EB using GEE multivariate analysis

Using the GEE multivariable model (Table 3), we identified seven significant risk factors for 3GCR-EB isolates in gestational urine cultures in our population: Arab ethnicity (OR, 1.22; 95% CI, 1.03–1.45), multiple (two or more) hospital admissions during the year before delivery (OR, 1.47; 95% CI, 1.21–1.79), AFP (OR, 1.53; 95% CI, 1.12–2.10), prescription of three and more antibiotic courses (OR, 1.76; 95% CI, 1.29–2.40), and specifically, usage of cephalosporins (OR, 1.56; 95% CI, 1.26–1.95), fluoroquinolones (OR, 1.34; 95% CI, 1.04–1.74), or nitrofurantoin (OR, 1.29; 95% CI, 1.02–1.64). Urinary tract anomalies and multiple fetus pregnancy covariates were non-significant in the GEE multivariable regression model.

**Table 1. Characteristics of pregnant women with EB growth in gestational urine cultures and univariate analysis for 3GCR-EB isolation.**

| Characteristics | Urine Culture Result | | | P value |
|---|---|---|---|---|
| | **Total EB-positive** | **EB-positive (non-3GCR-EB)** | **3GCR-EB-positive** | |
| | **n = 15,282** | **n = 14,679** | **n = 603** | |
| Age, y | | | | |
| < 20 | 382 (2.5%) | 368 (2.5%) | 14 (2.3%) | |
| 20–35 | 11,917 (78.0%) | 11,452 (78.0%) | 465 (77.1%) | 0.782 |
| > 35 | 2,983 (19.5%) | 2,859 (19.5%) | 124 (20.6%) | |
| SES | | | | |
| Low | 8,209 (53.7%) | 7,883 (53.7%) | 326 (54.1%) | |
| Medium | 4,685 (30.7%) | 4,503 (30.7%) | 182 (30.2%) | 0.967 |
| High | 2,388 (15.6%) | 2,293 (15.6%) | 95 (15.8%) | |
| Ethnic sector[a] | | | | |
| Jewish | 10,420 (68.2%) | 10,042 (68.4%) | 378 (62.7%) | |
| Arab | 4,862 (31.8%) | 4,637 (31.6%) | 225 (37.3%) | 0.004 |
| Multiple fetus pregnancy | 527 (3.4%) | 496 (3.4%) | 31 (5.1%) | 0.027 |
| AFP | 846 (5.5%) | 797 (5.4%) | 49 (8.1%) | 0.006 |
| Urinary tract anomalies | 182 (1.2%) | 166 (1.1%) | 16 (2.7%) | 0.001 |
| High-risk pregnancy | 1,891 (12.4%) | 1,824 (12.4%) | 67 (11.1%) | 0.369 |
| Diabetes Mellitus (non-gestational) | 790 (5.2%) | 748 (5.1%) | 42 (7.0%) | 0.053 |
| Gestational Diabetes Mellitus | 502 (3.3%) | 478 (3.3%) | 24 (4.0%) | 0.389 |
| Malignancy | 127 (0.8%) | 119 (0.8%) | 8 (1.3%) | 0.255 |
| Inflammatory Bowel Disease | 85 (0.6%) | 81 (0.6%) | 4 (0.7%) | 0.718 |
| Obesity | 2,546 (16.7%) | 2,449 (16.7%) | 97 (16.1%) | 0.741 |
| Hospital admission in the year before delivery | | | | |
| None | 8,277 (54.2%) | 8,005 (54.5%) | 272 (45.1%) | |
| One | 3,756 (24.6%) | 3,601 (24.5%) | 155 (25.7%) | <0.001 |
| More than one | 3,249 (21.3%) | 3,073 (20.9%) | 176 (29.2%) | |
| Antibiotics prescription during the pregnancy | | | | |
| Not prescribed | 5,356 (35.0%) | 5,225 (35.6%) | 131 (21.7%) | |
| Prescribed | 9,926 (65.0%) | 9,454 (64.4%) | 472 (78.3%) | <0.001 |
| Number of antibiotics courses dispensed | | | | |
| None | 5,356 (35.0%) | 5,225 (35.6%) | 131 (21.7%) | |
| One | 4,282 (28.0%) | 4,131 (28.1%) | 151 (25.0%) | <0.001 |
| Two | 2,531 (16.6%) | 2,435 (16.6%) | 96 (15.9%) | |
| Three or more | 3,113 (20.4%) | 2,888 (19.7%) | 225 (37.3%) | |
| Number of different types of antibiotics dispensed[b] | | | | |
| None | 5,356 (35.0%) | 5,225 (35.6%) | 131 (21.7%) | |
| One | 5,680 (37.2%) | 5,469 (37.3%) | 211 (35.0%) | <0.001 |
| Two | 3,033 (19.8%) | 2,877 (19.6%) | 156 (25.9%) | |
| Three or more | 1,213 (7.9%) | 1,108 (7.5%) | 105 (17.4%) | |
| Type of antibiotics | | | | |
| Cephalosporins | 5,787 (37.9%) | 5,452 (37.1%) | 335 (55.6%) | <0.001 |
| Penicillins | 5,439 (35.6%) | 5,175 (35.3%) | 264 (43.8%) | <0.001 |
| Nitrofurantoin | 1,508 (9.9%) | 1,410 (9.6%) | 98 (16.3%) | <0.001 |
| Fluoroquinolones | 1,167 (7.6%) | 1,088 (7.4%) | 79 (13.1%) | <0.001 |
| Macrolides | 858 (5.6%) | 822 (5.6%) | 36 (6.0%) | 0.767 |
| Tetracyclines | 557 (3.6%) | 528 (3.6%) | 29 (4.8%) | 0.148 |
| Sulfonamides | 160 (1.0%) | 148 (1.0%) | 12 (2.0%) | 0.034 |

*(Continued)*

**Table 1.** (Continued)

| Characteristics | Urine Culture Result | | | |
|---|---|---|---|---|
| | **Total EB-positive** | **EB-positive (non-3GCR-EB)** | **3GCR-EB-positive** | **P value** |
| | **n = 15,282** | **n = 14,679** | **n = 603** | |
| Aminoglycosides | 13 (0.1%) | 8 (0.1%) | 5 (0.8%) | <0.001 |
| Other | 184 (1.2%) | 173 (1.2%) | 11 (1.8%) | 0.217 |

Abbreviations: EB, enterobacteriaceae; 3GCR-EB, third-generation-cephalosporin resistant *Enterobacteriaceae*; y, years; SES, socioeconomic status; AFP, assisted fertilization procedure.

[a] Determined based on location of most-used community clinic.

[b] During the year prior to index date.

## Discussion

This study identified seven risk factors for 3GCR-EB, a surrogate for ESBL-PE, isolation in pregnant women. The proportion of 3GCR-EB in gestational urine cultures yielding EB in our

**Table 2. EB species in gestational urine cultures[a].**

| EB Species | EB-positive gestational urine cultures | | 3GCR-EB positive gestational urine cultures | |
|---|---|---|---|---|
| | **N = 15,439** | **100%** | **N = 611** | **%** |
| *Escherichia coli* | 11,515 | 74.6% | 437 | 71.5% |
| *Klebsiella pneumoniae* | 1,473 | 9.5% | 90 | 14.7% |
| *Proteus mirabilis* | 817 | 5.3% | 26 | 4.3% |
| *Klebsiella sp.* | 695 | 4.5% | 27 | 4.4% |
| *Citrobacter koseri* | 586 | 3.8% | 9 | 1.5% |
| *Morganella morganii* | 163 | 1.1% | 7 | 1.1% |
| *Klebsiella oxytoca* | 37 | 0.2% | 3 | 0.5% |
| *Klebsiella ornithinolytica* | 29 | 0.2% | 3 | 0.5% |
| *Serratia marcescens* | 21 | 0.1% | 2 | 0.3% |
| *Serratia fonticola* | 18 | 0.1% | 2 | 0.3% |
| *Escher coli Lact neg* | 14 | 0.1% | 1 | 0.2% |
| *Citrobacter sp.* | 13 | 0.1% | 0 | 0.0% |
| *Citrobacter freundii* | 13 | 0.1% | 2 | 0.3% |
| *Providencia stuartii* | 12 | 0.1% | 1 | 0.2% |
| *Citrobacter diversus* | 8 | 0.1% | 0 | 0.0% |
| *Klebsiella ozaenae* | 7 | 0.0% | 0 | 0.0% |
| *Proteus vulgaris* | 5 | 0.0% | 0 | 0.0% |
| *Providencia rettgeri* | 3 | 0.0% | 0 | 0.0% |
| *Hafnia alvei* | 1 | 0.0% | 0 | 0.0% |
| *Serratia rubidaea* | 1 | 0.0% | 0 | 0.0% |
| *Citrobacter sedlakii* | 1 | 0.0% | 0 | 0.0% |
| *Providencia alcalifaciens* | 1 | 0.0% | 0 | 0.0% |
| *Serratia sp.* | 1 | 0.0% | 0 | 0.0% |
| *Citrobacter amalonaticus* | 1 | 0.0% | 0 | 0.0% |
| Other (unidentified EB) | 4 | 0.0% | 1 | 0.2% |

Abbreviations: EB, *Enterobacteriaceae;* 3GCR-EB, third-generation-cephalosporin resistant *Enterobacteriaceae.*

[a] The total number of species is greater than the total number of positive cultures due to multiple EB species isolation in some cultures.

**Table 3. Risk factors for 3GCR-EB isolation in gestational urine cultures (GEE Regression).**

| Characteristics | OR | 95% CI | P value |
|---|---|---|---|
| Ethnic sector (ref: Jewish) | | | |
| Arab | 1.22 | (1.03–1.45) | 0.024 |
| AFP | 1.53 | (1.12–2.10) | 0.007 |
| Number of hospital admissions prior to delivery (ref: none) | | | |
| One | 1.20 | (0.98–1.47) | 0.075 |
| Two or more | 1.47 | (1.21–1.79) | <0.001 |
| Number of courses of antibiotics dispensed (ref: none) | | | |
| One | 1.12 | (0.86–1.46) | 0.406 |
| Two | 1.05 | (0.76–1.45) | 0.770 |
| Three or more | 1.76 | (1.29–2.40) | <0.001 |
| Type of antibiotics | | | |
| Cephalosporins | 1.56 | (1.26–1.95) | <0.001 |
| Fluoroquinolones | 1.34 | (1.04–1.74) | 0.024 |
| Nitrofurantoin | 1.29 | (1.02–1.64) | 0.036 |

Abbreviations: 3GCR-EB, third-generation-cephalosporin resistant *Enterobacteriaceae*; GEE, Generalized Estimating Equations; OR, odds ratio; 95% CI, 95% confidence interval; AFP, assisted fertilization procedure.

study was relatively low (3.9%), comparable with the ESBL-PE rates reported in a Norwegian study (2.9%)[25] and similar to the rates of ESBL-PE in Israeli study of urinary tract infections (UTIs) in a pediatric population (3.8%)[26]. In a recent study from Israel, the rate of ESBL-PE in women whose neonates were admitted to a NICU (as assessed by rectal screening) was significantly higher (21.5%) than the rate of ESBL-PE colonization that were derived just from gestational urine cultures in our study [15]. The rate of ESBL-PE in urine cultures is expected to be lower than the incidence assessed by direct selective screening, since other factors such as relative abundance of ESBL-PE in feces, are linked to the presence of these species as the ultimate causative pathogen from clinical samples [27].

Since detection of 3GCR-EB in urine usually represents fecal carriage of these species [23], the risk factors for 3GCR-EB (mostly ESBL-PE) isolation in gestational urine culture should translate into risk for ESBL-PE carrier state in pregnant women without gestational bacteriuria as well. However, this generalization could be confounded by factors that predispose for gestational bacteriuria itself [20].

The risk factors identified to be associated with resistant gram-negative bacteria in different populations can be generally categorized as health-care associated (co-morbidities, hospitalizations, etc.), antibiotic treatment–associated and other factors, usually connected to environmental exposures [19–22]. Among healthcare-associated factors in our study, we identified pregnancy achieved by AFP to be associated with 3GCR-EB. AFP as a specific risk factor for 3GCR-EB isolation has not been previously described in this context to the best of our knowledge. Despite the fact that AFPs are usually provided on an ambulatory basis, AFP clinics are often located in hospitals. We assume that healthcare exposure of women treated in these clinics is significant and comparable to recurrent hospitalizations, regarding the risk of ESBL-PE acquisition.

Previous hospitalizations and co-morbidities such as urinary tract abnormalities and diabetes have been well described in epidemiological studies of ESBL-PE in the general population [19]. Previous epidemiologic studies of ESBL-PE in pregnant women [13, 14] did not assess the impact of comorbidities as possible risk factors. In our cohort, we found multiple hospital

admissions as an important risk factor of ESBL-PE. Urinary tract abnormalities were also found to be a significant factor in the univariate analysis but not in the logistic regression. There was a trend for higher rates of 3GCR-EB isolation in women with diabetes but this finding did not reach statistical significance. We hypothesize that in this relatively healthy population of pregnant women, the impact of comorbidities is lower than in the general population and maybe an even larger sample is needed to demonstrate the association.

Antibiotic use is a well-described risk factor for infection by ESBL-PE [20, 21], though there are differences in the impacts attributed to different antibiotic groups. A study from Norway found that previous fluoroquinolone use was strongly associated with ESBL-PE infection [20]. A study from Syria, where the prevalence of ESBL-producing *E. coli* in UTIs exceeded 52%, also identified previous exposure to fluoroquinolones as a significant risk factor [21]. Similarly, in our study, fluoroquinolones were identified as a risk factor for 3GCR-EB isolation. Fluoroquinolones, classified as class C for pregnancy, are seldom used in gestation, unless there is no alternative therapy [28]. The contribution of fluoroquinolones to 3GCR-EB isolation despite this restricted usage emphasizes their relative importance for ESBL-PE selection [22]. Among the different antibiotic types, cephalosporins had the most significant association with ESBL-PE isolation as found in other studies [21]. Our study, in similar to recent publication [29], suggests that nitrofurantoin use is a risk factor for 3GCR-EB isolation. Unlike fluoroquinolones and cephalosporins, nitrofurantoin usage has been reported less frequently as a risk for ESBL-PE isolation in previous studies.

As expected, the risk for 3GCR-EB was quantitatively affected by antibiotic usage. Having three or more antibiotic courses was associated with the greatest risk.

Interestingly, isolation of 3GCR-EB was found to be significantly associated with Arab ethnicity, a finding that was described previously in Israeli children with UTIs, but not by other studies in the adult population in Israel [22]. The reason for the difference in 3GCR-EB rates between the two main Israeli ethnic sectors, Jewish and Arab, is currently unknown. The distinction between these populations in this regard can be attributed to differences in various environmental exposures and or patterns of antibiotic usage, but further studies are required to determine its possible causes.

The importance and strength of this study is its focus on third generation cephalosporin resistant carbapenem sensitive EB, mainly ESBL-PE in pregnant women, a subject that is not well characterized. Pregnant women are generally healthy, and thus are expected to have low rates of carrier state of resistant bacteria challenging prediction of this risk. Our access to a relatively large number of cases through the EHR database enables us to examine this population-based cohort of patients and to identify risk factors for resistant EB.

This study has several limitations. First, due to the study design we included for our analysis we did not distinguished between ESBL-PE and AmpC-producing bacteria. Nevertheless, while the AmpC mechanism for resistance to 3GC was described in all EB species, it is mostly characteristic of *Enterobacter* sp., *Citrobacter* sp., *Hafnia alvei*, *Morganella morganii*, *Serratia marcescens* and *Providencia* sp.[30]. These typical AmpC producers together, constitute only a minor fraction (4%, Table 2) in this study and in some of them, a mechanism of ESBL-PE could still account for cephalosporin resistance. Furthermore, infection control and antimicrobial therapy for ESBL-PE and AmpC producers are comparable.

Second, although the cohort was recruited from a large integrated health fund and the types of data used are similar to those used by other healthcare systems, our results are less useful in settings where outpatient clinics and hospital data are not linked. However, with the growing use of EHRs, the relevant data from community clinics may be increasingly available in many inpatient healthcare facilities.

In conclusion, the risk factors identified by this study for 3GCR-EB as a surrogate for ESBL-PE, in gestation, can be easily generalized for pregnant women in the Israeli population. Moreover, these risk factors, other than ethnicity, are applicable to pregnant women world-wide. Information of previous antibiotic treatments and hospitalization in the last year and AFPs are easily accessed and can be used for appropriate infection control practices and anti-microbial therapy.

## Supporting information

**S1 File. STROBE Checklist.**
(DOCX)

## Acknowledgments

The authors would like to thank Carly Davis-Pask, MPH (Clalit Research Institute, Israel) for her assistance in editing and reviewing the manuscript.

## Author Contributions

**Conceptualization:** Alex Guri, Natalie Flaks-Manov, Orna Flidel Rimon, Pnina Ciobotaro, Oren Zimhony.

**Data curation:** Alex Guri, Natalie Flaks-Manov, Adi Ghilai, Moshe Hoshen.

**Formal analysis:** Alex Guri, Natalie Flaks-Manov, Adi Ghilai, Moshe Hoshen.

**Investigation:** Alex Guri, Natalie Flaks-Manov, Oren Zimhony.

**Methodology:** Alex Guri, Natalie Flaks-Manov, Pnina Ciobotaro, Oren Zimhony.

**Project administration:** Alex Guri, Oren Zimhony.

**Software:** Adi Ghilai.

**Supervision:** Oren Zimhony.

**Validation:** Alex Guri, Moshe Hoshen.

**Writing – original draft:** Alex Guri, Natalie Flaks-Manov, Oren Zimhony.

**Writing – review & editing:** Alex Guri, Natalie Flaks-Manov, Adi Ghilai, Moshe Hoshen, Orna Flidel Rimon, Pnina Ciobotaro, Oren Zimhony.

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
