## [Decision Letter · Decision Letter 0]

2 Dec 2019

Risk factors for third-generation cephalosporin resistant Enterobacteriaceae in gestational urine cultures: A retrospective cohort study based on centralized electronic health records.

PONE-D-19-22821

Dear Dr. Guri,

We are pleased to inform you that your manuscript has been judged scientifically suitable for publication and will be formally accepted for publication once it complies with all outstanding technical requirements.

With kind regards,

Grzegorz Woźniakowski, PhD ScD

Academic Editor

PLOS ONE

Additional Editor Comments (optional):

Reviewers' comments:

Reviewer's Responses to Questions

**Comments to the Author**

1. Is the manuscript technically sound, and do the data support the conclusions?

Reviewer #1: Yes

2. Has the statistical analysis been performed appropriately and rigorously? 

Reviewer #1: Yes

3. Have the authors made all data underlying the findings in their manuscript fully available?

Reviewer #1: Yes

4. Is the manuscript presented in an intelligible fashion and written in standard English?

Reviewer #1: Yes

5. Review Comments to the Author

Reviewer #1: The manuscript represents a high level of science. Appropriately selected statistical methods, properly analyzed data. The discussion properly compares the results with the literature data. The conclusions were correctly formulated, as were the limitations.

6. PLOS authors have the option to publish the peer review history of their article (what does this mean?). If published, this will include your full peer review and any attached files.

Reviewer #1: Yes: Adam Kuzdraliński

---

## [Editor Report · Acceptance letter]

16 Dec 2019

PONE-D-19-22821 

Risk factors for third-generation cephalosporin resistant *Enterobacteriaceae* in gestational urine cultures: A retrospective cohort study based on centralized electronic health records. 

Dear Dr. Guri:

I am pleased to inform you that your manuscript has been deemed suitable for publication in PLOS ONE. Congratulations! Your manuscript is now with our production department. 

With kind regards,

on behalf of

Prof. Grzegorz Woźniakowski 

Academic Editor

PLOS ONE